# Galactooligosaccharides Promote Gut Barrier Integrity and Exert Anti-Inflammatory Effects in DSS-Induced Colitis Through Microbiota Modulation

**DOI:** 10.3390/ijms26167968

**Published:** 2025-08-18

**Authors:** Lucila A. Godínez-Méndez, Alejandra Natali Vega-Magaña, Marcela Peña-Rodríguez, Gisela Anay Valencia-Hernández, Germán Muñoz-Sánchez, Liliana Iñiguez-Gutiérrez, Rocío López-Roa, Martha Eloisa Ramos-Márquez, Mary Fafutis-Morris, Vidal Delgado-Rizo

**Affiliations:** 1Centro de Investigación en Inmunología y Dermatología (CIINDE), Departamento de Fisiología, Centro Universitario de Ciencias de la Salud, Universidad de Guadalajara, Guadalajara 44340, Mexico; lucila.godinez@edu.uag.mx (L.A.G.-M.); gisela.valencia6103@alumnos.udg.mx (G.A.V.-H.); gervvan.bio@gmail.com (G.M.-S.); mary.fafutis@academicos.udg.mx (M.F.-M.); 2Departamento Académico de Ciencias Básicas, Universidad Autónoma de Guadalajara, Av. Patria 120, Zapopan 45129, Mexico; liliana.gutierrez@edu.uag.mx; 3Laboratorio de Diagnóstico en Enfermedades Emergentes y Reemergentes (LADEER), Centro Universitario de Ciencias de la Salud, Universidad de Guadalajara, Guadalajara 44340, Mexico; alejandra.vega@academicos.udg.mx (A.N.V.-M.); marcela.pena@cucs.udg.mx (M.P.-R.); 4Laboratorio de Investigación en Cáncer e Infecciones (LICI), Centro Universitario de Ciencias de la Salud, Universidad de Guadalajara, Guadalajara 44340, Mexico; 5Laboratorio de Investigación y Desarrollo Farmacéutico (LIDF), Departamento de Farmacobiología, Centro Universitario de Ciencias Exactas e Ingenierías, Universidad de Guadalajara, Guadalajara 44430, Mexico; rocio.lopez@academicos.udg.mx; 6Departamento de Biología Molecular y Genómica, Centro Universitario de Ciencias de la Salud, Universidad de Guadalajara, Guadalajara 44340, Mexico; eloisa.ramos@academicos.udg.mx

**Keywords:** microbiota, ulcerative colitis, MCT1, MCT4, galactooligosaccharides, *Lupinus*

## Abstract

Ulcerative colitis is a chronic inflammatory bowel disease characterized by persistent inflammation, immune dysregulation, gut microbiota alterations, and impaired epithelial barrier function. *Lupinus albus* is a legume rich in galactooligosaccharides (GOS) that functions as a prebiotic capable of modulating the gut microbiota and mitigating ulcerative colitis-related damage. This study aimed to elucidate the effect of GOS on gut microbiota modulation and the molecular mechanisms involved in epithelial restoration and inflammation reduction. Fifteen C57BL/6 mice were randomly assigned to three groups (*n* = 5 per group): control (CTL), ulcerative colitis (UC), and ulcerative colitis + GOS (UC + GOS). UC was induced by administering 2% dextran sulfate sodium (DSS) in drinking water for seven days. The UC + GOS group received 2.5 g/kg BW of GOS via gavage for 14 days. GOS administration improved mucus layer thickness, regulated the expression of tight junction proteins, reduced pro-inflammatory cytokine levels, and modulated the gut microbiota, preventing the loss of richness and diversity. Additionally, the expression of monocarboxylate transporters (MCTs) MCT1 and MCT4 was evaluated, and significant differences were observed between the groups across colon and cecum tissues. These findings suggest that GOS supplementation may play a potential role in attenuating ulcerative colitis by regulating the gut microbiota and the metabolic state of intestinal cells.

## 1. Introduction

Ulcerative colitis (UC) is a chronic idiopathic inflammatory bowel disease (IBD) characterized by continuous mucosal inflammation that begins in the rectum and extends proximally through the colon [1]. The global incidence and prevalence of UC are rising, with an estimated 5 million cases reported worldwide in 2023 [2]. The disease is progressive and associated with several complications, such as gut dysmotility, hospitalization, anorectal disorders, colorectal cancer, and impaired quality of life [3]. Although the exact pathogenesis of UC remains unclear, it is understood to involve a complex interplay of factors, including a dysregulated immune response, gut microbiota alterations, genetic predisposition, and environmental influences [3].

One of the earliest and most critical alterations in UC is the disruption of the intestinal mucus barrier [4]. This protective layer serves as the first line of defense by limiting direct contact between luminal microbes and the host tissue and also functions as a reservoir for antimicrobial molecules that contribute to immune regulation and epithelial homeostasis [5]. When this barrier is compromised—as commonly observed in UC—the mucosa becomes more susceptible to bacterial invasion [6]. This facilitates the translocation of microbial products into the lamina propria, where they activate aberrant immune responses characterized by excessive production of pro-inflammatory cytokines such as IL-17, IL-6, IL-1β, and TNF-α [6,7]. These mediators impair mucosal healing, enhance immune cell infiltration, and perpetuate chronic inflammation [7]. Additionally, UC is associated with altered expression of tight junction proteins, including occludin, claudins, junction adhesion molecules (JAMs), and zonula occludens (ZOs), which further contribute to increased intestinal permeability and loss of epithelial integrity [8,9,10].

Closely linked to these barrier alterations is the role of the gut microbiota, a complex ecosystem composed primarily of bacteria, along with archaea, fungi, and viruses [11]. This microbial community is essential for maintaining intestinal immune homeostasis and supporting epithelial function [12]. In patients and murine models of UC, microbial dysbiosis is characterized by a reduction in microbial diversity and an imbalance between healthy and harmful bacteria [13]. Such dysregulation contributes to further barrier dysfunction, enhances mucosal permeability, and sustains intestinal inflammation [14].

One of the hallmark features of dysbiosis in UC is the reduction in beneficial short-chain fatty acid (SCFA)-producing bacteria, such as *Bifidobacterium longum*, *Eubacterium rectale*, *Faecalibacterium prausnitzii*, and *Roseburia intestinalis* [15]. These species play a key role in maintaining colonic health by producing SCFAs—particularly butyrate—that support mucus barrier function, reduce epithelial permeability, and suppress inflammatory signaling [15,16]. Patients with UC often exhibit an overabundance of harmful bacteria, particularly from the Proteobacteria phylum. Similar microbial shifts have been observed in dextran sulfate sodium (DSS)-induced colitis models, where Proteobacteria overgrowth exacerbates epithelial dysfunction and colonic inflammation [13,15].

SCFAs such as acetate, propionate, and butyrate are essential for maintaining colonic homeostasis [16]. Beyond their anti-inflammatory properties, they serve as an energy substrate for colonocytes and modulate immune responses [17]. Their cellular uptake is primarily mediated by monocarboxylate transporters (MCTs), particularly MCT1 and MCT4, which facilitate the transmembrane transport of SCFAs, lactate, and ketone bodies, contributing to epithelial energy balance [18]. Impaired MCT expression has been implicated in UC pathogenesis, further linking microbial dysbiosis to epithelial dysfunction [19].

Given this intricate relationship between gut microbiota and intestinal inflammation, modulation of the gut microbiota has emerged as a promising therapeutic strategy for UC [20]. Among various approaches, the use of prebiotics—particularly fermentable oligosaccharides—has gained attention due to their capacity to stimulate the growth of beneficial bacteria and enhance SCFA production [21,22]. Galactooligosaccharides (GOS) have demonstrated multiple health-promoting effects, including the stimulation of SCFA synthesis, promotion of epithelial cell growth and differentiation, regulation of host metabolism, and suppression of pathogenic bacterial overgrowth [23,24]. Experimental studies have shown that GOS supplementation can attenuate colonic inflammation, reduce levels of IL-1β and TNF-α, and increase levels of anti-inflammatory cytokines such as IL-10 [25,26]. However, the mechanism underlying these effects, especially those related to epithelial integrity and microbial–host interactions, remains to be fully elucidated.

This study aims to further investigate the effects of GOS derived from *Lupinus albus* in a murine model of UC, with particular emphasis on gut barrier integrity and anti-inflammatory outcomes mediated through gut microbiota modulation.

## 2. Results

### 2.1. GOS from Lupinus albus Attenuates the Damage of DSS Ulcerative Colitis

To assess the induction of ulcerative colitis, the disease activity index (DAI), body weight change, and colon length were evaluated. The UC group exhibited a significant increase in disease severity, as indicated by the DAI, a reduction in colonic tissue length, and body weight loss. In contrast, GOS administration mitigated disease severity, resulting in a lower DAI, preservation of colonic tissue length (published results) [27], and maintenance of body weight in mice (Figure 1A–D).

To evaluate mucus layer thickness, Alcian Blue staining with periodic acid–Schiff (PAS) counterstaining was performed. GOS treatment significantly improved mucus layer integrity (27.08 μm ± 4.21), maintaining a thickness approximately 52% greater than that observed in the UC group (13.09 μm ± 1.54) (Figure 1E,G).

Histological analysis was conducted to assess colonic inflammation. The UC group exhibited marked inflammatory cell filtration, with confluent leukocyte accumulation in the submucosa, complete crypt loss, extensive ulceration, and severe fibrosis (13.2 ± 3.96). In contrast, the UC + GOS group (7.8 ± 2.05) showed a significant 40% reduction in chronic inflammation in the descending colon tissue compared to the UC group (Figure 1F,H).

### 2.2. Modification of the Gut Microbiota by the DSS-Induced Ulcerative Colitis and GOS Administration

To determine whether GOS supplementation prevents gut microbiota dysbiosis in the DSS-induced UC model, we compared the microbial composition across the experimental groups. Microbial richness (Chao index) and diversity (Shannon index) were assessed. On day 14, both richness and diversity were significantly reduced in the UC group compared with the CTL and UC + GOS groups (Figure 2A,B). These findings indicate that DSS-induced colitis diminishes microbiota richness and diversity, whereas GOS treatment preserves these parameters.

To further investigate the structural differences in microbial communities between groups, non-metric multidimensional scaling (NMDS) analysis was performed. NMDS allows for the visualization of microbial similarity and dissimilarity by clustering samples based on their composition. The results revealed distinct microbial structures among the experimental groups, with significant separation observed (PERMANOVA F = 3.6116, R^2^ = 0.27, *p* = 0.001) (Figure 2C).

To gain a deeper insight into the specific taxonomic shifts associated with UC and GOS treatment, the relative abundance of bacterial phyla across the experimental groups was examined. A significant reduction in the Bacteroidetes phylum and an increase in the Firmicutes phylum were detected in the UC group (Figure 2D,E). In contrast, the UC + GOS group also exhibited a reduction in Bacteroidetes but preserved a higher relative abundance of this phylum compared to the UC group, suggesting a partial protective effect of GOS supplementation.

To further characterize microbiota alterations at a finer taxonomic resolution, bacterial family composition was analyzed. On day 14, the UC group showed a significant increase in the relative abundance of Streptococcaceae, Micrococcaceae, and Actinomycetaceae, accompanied by a marked decrease in Prevotellaceae and Fusobacteriaceae abundance compared to the CTL group (Figure 2F,G). In contrast, GOS administration mitigated these alterations, maintaining a bacterial profile at the family level that closely resembled that of the CTL group. The only notable difference in the UC + GOS group was an increase in Lachnospiraceae relative to the UC group (Figure 2G).

At the genus level, distinct changes were observed in response to UC induction and GOS treatment. On day 14, the UC group exhibited a significant reduction in the relative abundance of *Prevotella, Fusobacterium*, *Oribacterium*, and *Alloprevotella*, along with an increase in *Streptococcus* and *Actinomyces* compared to the CTL group (Figure 2H,I). In contrast, the UC + GOS group displayed a significant reduction only in *Prevotella*. Importantly, GOS administration preserved the abundance of *Fusobacterium*, *Oribacterium,* and *Alloprevotella* at levels comparable to the CTL group (Figure 2I). These findings highlight the protective role of GOS in modulating microbial composition and preventing excessive shifts in key microbial taxa associated with gut health.

Finally, to assess long-term microbial alterations associated with UC and the potential protective effects of GOS, we analyzed microbiota composition between baseline and post-colitis states. Our results showed a significant decline in the microbial richness and diversity in the UC group, as reflected by the Chao and Shannon indices, respectively (Appendix A). In contrast, both UC + GOS and CTL groups maintained stable alpha diversity levels between basal and day 14 timepoints, suggesting that GOS supplementation helps preserve microbial diversity under inflammatory conditions (Appendix A).

To complete these findings, we assessed the beta diversity through NMDS analysis. No significant differences were found between the pre- and post-colitis states in either the UC group (PERMANOVA F = 1.7946, R^2^ = 0.11 *p* = 0.13) or the UC + GOS group (PERMANOVA F = 1.698, R^2^ = 0.11, *p* = 0.11) (Appendix A).

To further explore taxonomic changes, we compared the relative abundance of bacterial groups before and after colitis induction. In the UC group, a significant increase in the abundance of the phylum Actinomycetota and a reduction in the abundance of Bacteroidetes were observed (Appendix A). Conversely, the UC + GOS group exhibited a taxonomic profile similar to that of the CTL group, with Proteobacteria being the only phylum showing a notable increase (Appendix A).

At the family level, the UC group showed a significant increase in the abundance of Micrococcaceae and Actinomycetaceae, along with decreases in the abundance of Prevotellaceae and Fusobacteriaceae (Appendix A). Meanwhile, in the UC + GOS group, only the abundance of Micrococcaceae was significantly elevated (Appendix A).

Genus-level analysis revealed that the UC group had a significant reduction in the abundance of *Prevotella, Fusobacterium, Alloprevotella*, and *Leptotrichia* and increases in the abundance of *Rothia* and *Actinomyces* (Appendix A). Similarly, the UC + GOS group showed a decrease in the abundance of *Prevotella* and a trend toward increased *Rothia* abundance, mirroring the pattern observed in the UC group (Appendix A). However, additional shifts were observed in the UC + GOS group, including reductions in *Kingella, Porphyromonas*, and *Selenomonas*, suggesting a selective modulation of microbial populations (Appendix A).

Since an increase in the abundance of Proteobacteria phylum was observed in the UC + GOS group, we further analyzed the genera *Neisseria* and *Haemophilus*, which displayed a trend toward increased abundance, although the differences were not statistically significant (Appendix A). We also noted a tendency toward increased *Lactobacillus* abundance; nevertheless, this increase did not reach statistical significance (Appendix A).

These findings suggest that GOS administration helps preserve microbial richness and diversity in the context of UC while selectively modulating specific bacterial taxa, potentially contributing to gut homeostasis.

### 2.3. DSS-Induced Ulcerative Colitis Modifies Cecum Microbiota Diversity

Considering the cecum as a key site of microbial fermentation, we analyzed its microbiota composition to assess the effect of GOS administration and DSS-induced colitis. Our results showed that microbial richness, as measured through the Chao index, was not significantly affected by either UC induction or GOS treatment (Figure 3A). However, microbial diversity, assessed through the Shannon index, was significantly reduced in the UC group compared to the UC + GOS group (Figure 3B). Beta diversity analysis using NMDS revealed no distinct clustering of microbial community structures among the groups (PERMANOVA F = 0.8773, R^2^ = 0.77, *p* = 0.54) (Figure 3C).

At the taxonomic level, the UC group exhibited an increased abundance of the phylum Proteobacteria, driven by elevated levels of the Pasteurellaceae family and the *Haemophilus* genus (Figure 3D–G). In contrast, the UC + GOS group maintained a microbial composition comparable to that of the CTL group, without significant alterations in taxonomic distribution (Figure 3D–G).

### 2.4. Comparison of Gut Microbiota Composition Between 14-Day and Cecum Feces

To continue evaluating the gut microbiota composition, we examined whether the UC model and GOS treatment influenced differences between fecal samples collected on day 14 and those from the cecum across experimental groups. Microbial richness and diversity were assessed at both sampling sites within each group, and no significant differences were detected in either index (Figure 4A,B).

To evaluate the structural distribution of microbial communities, NMDS analysis was conducted comparing day 14 feces with cecum feces within each group (Figure 4C–E). Notably, significant differences were observed only in the CTL group, where a distinct separation between day 14 feces and cecum feces samples was identified (PERMANOVA F = 3.6597, R^2^ = 0.21968, *p* = 0.016; Figure 4C).

Additionally, we analyzed overall similarity and dissimilarity by pooling both sample types (day 14 and cecum) and filtering by experimental group (Appendix A). A final comparison was performed by grouping all experimental conditions and filtering by sampling sites (Appendix A). These broader comparisons did not reveal statistically significant differences.

To gain further insight into the site-specific taxonomic composition, we analyzed the relative abundance of bacterial taxa at the phylum, family, and genus levels. At the phylum level, the CTL group exhibited marked differences, with significantly higher proportions of Proteobacteria and Bacteroidetes in day 14 feces compared to cecum feces (Figure 4F). Although the UC group did not exhibit statistically significant changes, a higher proportion of Proteobacteria was observed in cecum feces relative to day 14 feces (Figure 4F). In contrast, the UC + GOS group maintained a phylum-level composition comparable to the CTL group (Figure 4F).

At the family level, the CTL group displayed greater microbial abundance in cecum feces, including a significant increase in Leptotrichiaceae and Flavobacteriaceae (*p* ≤ 0.05) (Figure 4G). Additionally, families such as Streptoccocaceae, Micrococcaceae, Porphyromonadaceae, Actinomycetaceae, and Lactobacillaceae were also more abundant in the cecum feces (Figure 4G).

At the genus level, the CTL group exhibited a significantly higher proportion of *Leptotrichia* and a lower proportion of *Prevotella* in cecum feces compared to day 14 feces (Figure 4H). The UC group displayed an increased abundance of the Pasteurellaceae family, along with enrichment of *Haemophilus* and *Actinobacillus* in cecum feces relative to day 14 feces (Figure 4G,H). Conversely, the UC + GOS group showed minimal differences between sampling sites, with only a higher abundance of Leptotrichiaceae and *Leptotrichia* in the cecum (Figure 4G,H).

These findings suggest that the CTL group exhibited greater microbial variability and compartmentalization in the cecum, likely reflecting physiological differentiation between intestinal regions. In contrast, ulcerative colitis disrupted this pattern, reducing microbial diversity and promoting the overgrowth of potentially harmful bacteria. Notably, GOS administration helped maintain a stable microbiota composition across intestinal sites, resembling that of the CTL group and preventing excessive expansion of Proteobacteria in the cecum.

### 2.5. DSS-Induced Ulcerative Colitis Alters Monocarboxylate Transporter Expression

Building upon the observed shifts in microbial composition, we next assessed the expression of monocarboxylate transporters to elucidate how epithelial damage induced by the UC may alter SCFA transport dynamics. Under normal physiological conditions (CTL group), both MCT1 and MCT4 were significantly more expressed in the cecum compared to the descending colon. Specifically, MCT1 expression in the cecum was three times higher (7.21 ± 1.47) than in the descending colon (2.39 ± 1.23), while MCT4 expression in the cecum was six-fold greater (3.86 ± 1.14) than in the descending colon (0.61 ± 0.36) (Figure 5A,B).

In contrast, the UC group showed a marked reduction in MCT1 expression in the cecum (3.5 ± 1.46), indicating potential impairment in SCFA uptake. However, animals treated with GOS (UC + GOS group) preserved MCT1 expression levels (5.13 ± 2.7), corresponding to a 32% increase relative to the UC group. Although differences in MCT1 expression in the descending colon tissue were not statistically significant, the UC group (0.98 ± 0.62) showed a downward trend, with expression levels 60% lower than those in the CTL group. The UC + GOS group (1.27 ± 0.44) maintained MCT1 expression approximately 30% higher than the UC group (Figure 5A).

Regarding MCT4 expression, no statistically significant differences were observed in the cecum. However, the UC group (1.29 ± 0.22) exhibited a 67% reduction compared to the CTL group (3.86 ± 1.14), while the UC + GOS group (2.64 ± 1.05) showed a partial recovery, with levels 50% higher than those observed in the UC group. Conversely, in the descending colon, UC triggered a substantial increase in MCT4 expression (6.76 ± 6.36), reaching levels 11 times greater than in the CTL group (0.61 ± 0.36), possibly as a compensatory response to inflammation. Notably, MCT4 expression in the UC + GOS group (1.6 ± 2.5) was reduced compared to the UC group, aligning more closely with the CTL group (Figure 5B).

To better understand the functional significance of MCT expression, we analyzed its correlation with SCFA concentration in the cecum. In the CTL group, MCT1 expression showed a strong positive correlation with butyrate levels (R^2^ = 0.923, *p* = 0.01) (Appendix A). While no significant correlation was found in the UC group, a trend was observed with propionic acid levels (R^2^ = 0.725, *p* = 0.0673) (Appendix A). Interestingly, in the UC + GOS group, MCT1 expression was significantly correlated with acetic acid levels (R^2^ = 0.9351, *p* = 0.01) (Appendix A).

Taken together, these findings suggest that MCT expression could be influenced by both inflammatory status and SCFA availability, potentially affecting epithelial homeostasis. The ability of GOS to preserve MCT expression and its association with SCFA levels underscore its role in supporting mucosal health and mitigating UC-related damage.

### 2.6. GOS Administration Reduces the Expression of Inflammatory Cytokines in Descending Colonic Tissue

To explore the immunomodulatory effects of GOS treatment, we evaluated the expression of key cytokines in the descending colonic tissue. As expected, induction of colitis with DSS triggered a strong pro-inflammatory response, evidenced by significant upregulation of TNF-α, IL-6, and IL-17 expression. Compared to the CTL group, the UC group exhibited a 6-fold increase in TNF-α expression (CTL = 2.51 ± 2.16 vs. UC = 15.76 ± 4.7), a 47-fold increase in IL-6 expression (0.0003 ± 0.0002 vs. 0.014 ± 0.01), and a 30-fold increase in IL-17 expression (0.004 ± 0.003 vs. 0.12 ± 0.08), confirming the exacerbated inflammatory milieu associated with UC (Figure 6A–C).

Conversely, GOS administration attenuated this inflammatory response. TNF-α levels in the UC + GOS group (5.92 ± 7.14) were reduced by approximately 3-fold relative to the UC group. Similarly, IL-6 (0.0004 ± 0.0001) and IL-17 (0.019 ± 0.009) expression levels were significantly lower in the UC + GOS group, approaching baseline values observed in the CTL group (Figure 6A–C).

In addition to suppressing pro-inflammatory cytokines, GOS treatment also enhanced anti-inflammatory signaling. Notably, IL-10 expression was significantly elevated in the UC + GOS group (7.65 ± 4.88), representing a ~50-fold increase compared to the CTL group (0.15 ± 0.11). While the UC group also exhibited an increase in IL-10 expression (0.84 ± 0.57), this ~5-fold elevation relative to the CTL group appeared insufficient to counterbalance the intensified inflammatory response (Figure 6D).

These findings indicate that GOS treatment exerts a dual immunomodulatory effect by downregulating pro-inflammatory mediators and upregulating IL-10 expression, thereby contributing to the control of inflammation.

### 2.7. GOS Treatment Modulates Tight Junction Protein Expression in Descending Colonic Tissue

Disruption of the intestinal epithelial barrier is a hallmark of UC, primarily driven by alterations in TJ proteins that compromise mucosal integrity. To evaluate the impact of GOS on epithelial barrier function, we assessed the expression of six key TJ proteins in the descending colon.

Our analysis revealed that UC profoundly disrupted the expression profile of several TJ proteins, indicating significant epithelial barrier dysregulation. Compared to the CTL group, the UC group exhibited a ~6-fold increase in JAM-A expression (CTL = 5.03 ± 1.73 vs. UC = 31.77 ± 12.4), a 3-fold increase in ZO-1 expression ( 2.82 ± 1.55 vs. 6.98 ± 3.02), a 15-fold increase in claudin-2 expression (0.53 ± 0.36 vs. 8.17 ± 4.13), and a ~20-fold increase in claudin-4 expression (1.65 ± 1.5 vs. 32.6 ± 28.4), all consistent with a state of impaired mucosal barrier function (Figure 7A,C,E,F).

Remarkably, GOS administration preserved TJ protein expression at levels comparable to those observed in the CTL group. In the UC + GOS group, the expression levels of JAM-A (6.44 ± 4.6), ZO-1 (2.6 ± 1.5), claudin-2 (0.96 ± 1.3), and claudin-4 (9.16 ± 13.1) were significantly lower than in the UC group (Figure 7A,C,E,F).

Furthermore, GOS treatment tended to increase occludin expression, showing a 3-fold increase compared to the UC group (13.7 ± 5 vs. 4.5 ± 1.9, *p* = 0.06) and a 2-fold increase relative to the CTL group (7.86 ± 5.1) (Figure 7B). In contrast, no significant differences were observed in claudin-1 expression among the experimental groups (CTL = 0.15 ± 0.11, UC = 0.22 ± 0.23, UC + GOS = 0.1 ± 0.04) (Figure 7D).

These findings suggest that UC disrupts TJ protein expression, compromising intestinal barrier integrity, while GOS treatment attenuates this disruption. Thus, GOS may play a role in maintaining mucosal barrier function and potentially contribute to the regulation of intestinal inflammation in UC.

## 3. Discussion

This study explored the potential effect of *Lupinus albus*-derived galactooligosaccharides in a DSS-induced model of UC. *Lupinus albus*, commonly known as white lupin, is a legume increasingly recognized for its high protein content and potential pharmaceutical applications [28]. White lupin is rich in dietary fiber, accounting for approximately 40% of its composition, primarily composed of raffinose-family oligosaccharides (RFOs) [29]. These oligosaccharides have gained attention for their prebiotic properties, as they promote the growth of beneficial gut microbiota [23,24,30]. However, the specific effects of GOS derived from *Lupinus albus* on inflammatory conditions such as UC remain largely unexplored.

In our study, seven days of DSS treatment led to UC development, as evidenced by body weight loss, colon shortening, increased DAI, and a thinner mucus layer. These findings are consistent with previous research by Johansson et al., who demonstrated that DSS-induced colitis results in mucus layer depletion, permitting bacterial contact with epithelial cells and triggering severe inflammation [31]. Similarly, our UC group exhibited significant mucus loss. In contrast, GOS administration preserved mucus layer thickness, prevented weight loss, and reduced disease activity and inflammation scores. These results align with findings from Liu et al., who reported that a combination of GOS and other compounds also reduced inflammation and preserved the mucus barrier [32].

Given the critical role of the gut microbiota in intestinal homeostasis, we assessed microbial composition across experimental groups. Our results demonstrated that UC induction led to reduced microbial richness and diversity, promoting a dysbiotic environment characterized by loss of beneficial genera and enrichment of opportunistic bacteria. GOS treatment significantly mitigated these alterations. At the taxonomic level, both the UC and UC + GOS groups exhibited a decrease in the abundance of the Bacteroidetes phylum and its representative genus *Prevotella*; however, the UC + GOS group preserved higher levels of these taxa compared to the UC group. In contrast, the UC group showed a notable increase in the abundance of the Firmicutes phylum, mainly driven by the expansion of the *Streptococcus* genus. These findings are consistent with clinical observations, such as those reported by Kyung-Hyo et al., where UC patients had a higher relative abundance of Firmicutes and lower Bacteroidetes compared to healthy individuals [33]. Beyond these clinical results, the role of *Streptococcus* in UC pathogenesis has also been explored in experimental settings. *Streptococcus* species, commonly associated with the oral microbiota and dental caries, have been implicated in the exacerbation of colitis through oral–gut translocation [34,35]. For instance, administration of *Streptococcus mutans* in a DSS-induced colitis model worsened mucosal damage and inflammation, underscoring the pro-inflammatory capacity of specific bacterial species and their potential contribution to UC pathogenesis through immune system activation [36].

In our study, the UC group showed a significant reduction in the relative abundance of *Oribacterium*, while in the UC + GOS group, this genus was preserved. *Oribacterium* belongs to the Lachnospiraceae family, which is recognized for its SCFA-producing genera and its role in maintaining a beneficial gut microbiota composition [37]. Consistent with our findings, Matsumoto et al. reported a significant depletion of *Oribacterium* in UC patients, regardless of their response to 5-aminosalicylic acid (5-ASA) treatment, compared to healthy controls [38]. Although the role of the *Oribacterium* in UC remains poorly characterized, other models have shown that *Oribacterium* sp. GMBB0313 exerts an immunomodulatory effect, particularly in enhancing CD8+ T cell responses during SARS-CoV-2 infection [39]. These findings support the hypothesis that GOS selectively modulates microbiota composition, potentially counteracting UC-associated dysbiosis and offering immunomodulatory benefits.

Additionally, we evaluated the gut microbiota composition before and after colitis induction in each experimental group. As expected, the UC group showed a marked reduction in microbial richness and diversity following DSS administration. In contrast, the UC + GOS group maintained relatively stable microbial profiles between baseline and post-colitis conditions. NMDS analysis did not reveal significant intra-group differences over time. In the UC group, this may be due to the concentration and duration of DSS exposure, which may have been insufficient to induce consistent and detectable shifts across all animals. Similar findings were reported by Park et al., who evaluated microbial modulation in a DSS-induced colitis model using different DSS concentrations (1%, 2%, and 3%). In that study, significant microbial alterations were only observed with 2% and 3% DSS from day 8 onward [40].

In our model, colitis severity appeared to be influenced by baseline body weight and individual susceptibility. Notably, mice weighing less than 20 grams at baseline of the experiment exhibited more severe clinical manifestations, including reduced activity and increased mortality—findings consistent with previous reports [41,42,43]. Furthermore, in the UC group, the high relative abundance of the Firmicutes phylum (>60%) observed both at baseline and after post-colitis induction may have masked more subtle shifts in other bacterial groups, which became evident only when taxonomic composition was analyzed at the genus level.

In line with these findings, UC led to a loss of *Fusobacterium*, *Alloprevotella*, *Leptotrichia*, and *Prevotella* genera, along with an increase in the abundance of *Rothia* and *Actinomyces.* These results align with those described by Bajer et al., who found a similar microbial pattern in patients with sclerosing cholangitis and UC—characterized by an increased abundance of *Rothia* and a decrease in *Prevotella* [44]. The role of *Rothia* is controversial; although it is a normal component of the gut microbiota and contributes to homeostasis, its overgrowth has been associated with metabolic alterations and autoimmune inflammatory diseases, though the precise mechanisms are still unclear [45,46].

In contrast, GOS treatment maintained a more stable microbial profile, preventing most of the UC-associated alterations, except for a slight reduction in *Prevotella*. Interestingly, previous studies have shown that GOS supplementation can increase the abundance of *Prevotella* and other beneficial bacteria such as *Lactobacillus*, while reducing potentially harmful taxa. Supporting this, Nishihara et al. reported that patients with active UC who failed to respond to 5-ASA treatment or developed resistance exhibited a marked depletion of *Prevotella* [47]. In this context, the preservation of *Prevotella* observed with GOS intervention may help prevent its complete loss and could contribute to improved clinical outcomes, particularly in individuals with limited response to conventional therapies.

Another relevant microbial shift in the UC group was the increased abundance of *Actinomyces.* Although this genus is a commensal member of the oral and gut microbiota, its overgrowth has been associated with pathological conditions, particularly in immunocompromised individuals, where it can lead to abscess formation and progressive tissue damage [48,49,50]. Beyond these clinical manifestations, emerging evidence suggests a potential role of *Actinomyces* in immune modulation and tumorigenesis. Elevated levels of this genus in the gut have been shown to co-localize with cancer-associated fibroblasts (α-SMA+) and activate the TLR2/NF-κB signaling pathways, contributing to the reduction of CD8+ T cells in the colorectal cancer microenvironment [51]. In our study, GOS supplementation prevented the overgrowth of this genus, suggesting that its modulation may help reduce inflammatory disturbance. However, further studies are needed to clarify the underlying immunological mechanisms.

Notably, the UC + GOS group exhibited a reduction in the genera *Selenomonas*, *Kingella*, and *Porphyromonas,* along with a tendency towards increased abundance of *Neisseria, Haemophilus*, and *Lactobacillus*. The reduction in *Selenomonas*, *Kingella*, and *Porphyromonas* may positively contribute to gut microbial balance, as these genera are associated with lipopolysaccharide (LPS) production and inflammatory responses [52,53,54].

Given the observed increase in the Proteobacteria phylum in the UC + GOS group after colitis induction, we further examined the abundance of the *Haemophilus* genus. When comparing *Haemophilus* levels post-induction to baseline, we observed a nearly threefold increase in the UC + GOS group. Although a similar trend was noted in the CTL group between the basal and the day 14 timepoints. These findings suggest that the increase in *Haemophilus* abundance may be influenced by temporal or host-related factors unrelated to colitis induction or GOS intervention.

Species of the genus *Neisseria*, although often associated with pathogenic strains like *N. gonorrhoeae* and *N. meningitidis*, also comprise over ten commensal species that are part of the normal human microbiota [55]. For example, *Neisseria elongata* has been reported to inhibit the colonization of *N. gonorrhoeae* through the production of antimicrobial compounds. Similarly, *Neisseria mucosa,* a non-pathogenic species, has demonstrated the ability to kill *N. gonorrhoeae* in co-culture assays, suggesting a protective role within the microbial community [56]. Commensal *Neisseria* species produce compounds such as bacteriocins, which suppress the growth of competing bacteria by damaging membranes and degrading nucleic acids [57].

In this context, the increase in *Neisseria* observed following GOS treatment may contribute to maintaining mucosal homeostasis by limiting the colonization of potential pathogens and mitigating mucosal damage. However, further research is required to elucidate the mechanisms and functional implications of this microbial shift.

The increase in *Lactobacillus* species observed with GOS treatment is noteworthy, as these bacteria have been shown to ameliorate inflammation in UC by reinforcing the intestinal barrier, modulating the gut microbiota, and increasing SCFA production [58,59]. These findings suggest that GOS administration improves gut microbiota composition, counteracts dysbiosis associated with UC, and may serve as a complementary therapeutic strategy.

To gain further insight into the role of the microbiota in inflammation, we evaluated the gut microbial composition in the cecum, making this the first study to assess GOS treatment in UC using cecum feces. Our findings indicate that GOS improved cecum microbiota diversity, whereas the UC group exhibited a significant reduction. We also observed an increase in the proportion of the Proteobacteria phylum in the UC group, while the CTL and UC + GOS groups maintained lower levels. These results are consistent with those reported by Overstreet et al., who found elevated Proteobacteria levels in cecum samples from IL-10^−/−^ mice with spontaneous colitis [60].

We further compared the gut microbiota composition between the cecum and 14-day fecal samples. The patterns observed in the fecal samples mirrored those found in the cecum. NMDS analysis revealed significant differences in the CTL group, with higher variability in the cecum feces compared to 14-day feces. In contrast, the UC group displayed lower cecum microbial variability. GOS treatment preserved a microbial composition in the cecum similar to that of the CTL group.

Previous studies have reported structural differences between the cecum and colonic microbiota [61,62,63]. Although research specifically addressing these differences in UC models is limited, Lavelle et al. demonstrated that microbial composition varied depending on sample type (luminal vs. mucus-associated) in both healthy and UC patients, while no significant differences were observed between distinct colorectal regions (cecum, transverse, and descending colon) [64]. Taken together, these findings suggest that under healthy conditions, the cecum and descending colon harbor relatively distinct microbial dynamics. However, UC appears to disrupt this balance, promoting altered cecum microbiota composition with increased Proteobacteria, whereas GOS administration helps prevent these changes.

To assess the relationship between gut microbiota and epithelial barrier transport, we evaluated the expression of MCT1 and MCT4. MCTs are involved in cellular metabolic activity through the transport of energy substrates. MCT1 primarily facilitates the uptake of SCFAs, ketone bodies, pyruvate, and lactate, while MCT4 plays a crucial role in lactate efflux [65]. The transport of these energy substrates into cells is essential for maintaining tissue homeostasis [66]. In colonic tissue, MCT1 expression is higher in the cecum compared to the descending colon, and it is positively correlated with SCFA uptake, particularly butyrate [67].

Our study found a significant reduction in MCT1 expression in cecum tissue and a tendency for reduced expression in descending colonic tissue during UC. However, GOS administration preserved MCT1 expression in the cecum, maintaining levels 1.5-fold higher than those observed in the UC group. In the descending colon, GOS treatment did not significantly alter MCT1 expression compared to the UC group. These findings are consistent with the results reported by Ferrer et al., who evaluated the effect of butyrate on the intestinal epithelium in patients with Crohn’s disease (CD) and UC. They found that the expression of butyrate transporters, including MCT1 and ABCG2, was reduced in inflamed mucosa in both conditions and often undetectable [19].

Additionally, we investigated the correlation between SCFA production in cecum feces and MCT1 expression. In the CTL group, butyrate levels positively correlated with MCT1 expression. However, no such correlation was found in the UC group. In contrast, the UC + GOS group showed a positive correlation between acetic acid levels and MCT1 expression, as well as with total SCFA production. These results suggest that acetic acid may play a role in regulating the metabolic state of cells during inflammation. Supporting this, Deleu et al. found that high concentrations of acetate protect the intestinal barrier and exert anti-inflammatory effects, enhancing epithelial resistance in UC patients [68]. The proposed mechanism involves acetate metabolism via the tricarboxylic acid (TCA) cycle, which contributes to ATP production and prevents catabolic stress in intestinal cells [69]. Altogether, these findings suggest that GOS administration may enhance SCFA transport into colonic tissue and regulate cellular metabolism; however, further research is needed to fully confirm this hypothesis.

Regarding MCT4, which is often associated with tumor malignancies in various tissues, we found higher expression in cecum tissue than in descending colonic tissue under physiological conditions. In UC, MCT4 expression was upregulated in descending colonic tissue but downregulated in the cecum. In contrast, the UC + GOS group showed a 4-fold reduction in MCT4 expression in the descending colon compared to the UC group, while maintaining levels comparable to the CTL group in cecum tissue. MCT4 regulation has recently gained attention for its role in tumor progression across several types of cancers [70]. In colorectal cancer, MCT4 overexpression has been correlated with poor survival in rectal cancer patients [71]. Moreover, inhibition of MCT4 has been shown to improve T cell function and promote tumor cell lysis [70]. These findings suggest that GOS treatment may beneficially modulate MCT4 expression, potentially contributing to the prevention of UC-associated malignancies. Additionally, evaluating lactate levels is essential for fully understanding the variability in MCT4 expression between the cecum and descending tissues.

One of the major challenges in UC is the upregulation of inflammatory mediators, particularly pro-inflammatory cytokines. In our study, UC induced an increase in pro-inflammatory cytokines while simultaneously maintaining a lower concentration of IL-10. In contrast, GOS treatment resulted in reduced expression of pro-inflammatory cytokines and an increased proportion of IL-10. These findings may be related to alterations in gut microbiota composition and SCFA production, as previous studies have shown that elevated acetate concentrations (100 mM) can downregulate the expression of IL-1β, IL-2, IL-4, and IL-6 [68]. Moreover, GOS treatment appears to support microbial balance, enhancing the abundance of *Lactobacillus* and *Lachnospiraceae* species, which may contribute to the suppression of immune activation and inflammatory mediator expression [59,72].

To better understand the mechanism by which GOS improves colonic tissue health, we assessed the expression of key TJ proteins. Our results showed that the UC increased the expression of claudin-2 and claudin-4. This finding is consistent with the study by Cuzic et al., who evaluated claudin expression in both human samples and DSS-induced murine models of UC. They reported that, in mice, claudin-2 was overexpressed in the proliferative crypt zones, whereas claudin-4 was upregulated in lymphoid follicles and inflammatory cells within the lamina propria [8]. The overexpression of these claudins has been linked to epithelial barrier disruption and inflammation [73,74]. Notably, GOS treatment prevented the upregulation of both claudin-2 and claudin-4 in the descending colon, suggesting a protective role in preserving epithelial integrity.

Additionally, the GOS-treated group maintained higher levels of occludin expression, which is essential for maintaining barrier function and preventing epithelial damage [75]. Interestingly, we observed increased expression of JAM-A and ZO-1 in the UC group. However, these findings differ from previous reports [76,77]. JAM-A is involved in leukocyte diapedesis during inflammation and contributes to epithelial permeability regulation [78]. It has been shown that exposure of epithelial cells to pro-inflammatory cytokines such as TNF-α, IL-22, IFN-γ, and IL-17 leads to an increase in the expression of phosphorylated JAM-A (p-JAM-A) [79]. Elevated levels of p-JAM-A have been reported in both DSS-induced colitis models and UC patients, particularly in crypt epithelial cells [79]. Although we did not evaluate JAM-A phosphorylation in this study, the observed increase in total expression may contribute to tight junction dysfunction and barrier disruption.

ZO-1 is a scaffold protein that interacts with JAM-A and other transmembrane TJ components, linking them to the actomyosin cytoskeleton and mediating signal transduction [80]. ZO-1 plays a central role in maintaining TJ stability and epithelial architecture [77]. While ZO-1 is typically downregulated in active UC and DSS-induced murine models [81,82] some studies have reported elevated ZO-1 expression in quiescent UC patients and in certain cancers, including colorectal and bladder cancer [83,84]. In our study, the upregulation of ZO-1 in the UC group may reflect an inflammatory milieu and its potential interaction with JAM-A during leucocyte transmigration, ultimately contributing to epithelial barrier dysfunction.

Considering all of these results, it can be concluded that GOS administration improves intestinal health by preserving the thickness of the mucus layer, likely through the promotion of a beneficial gut microbiota. Additionally, GOS appears to support cellular metabolic activity by enhancing the transport of SCFAs into colonic cells. This treatment also contributes to the downregulation of pro-inflammatory cytokines and the upregulation of IL-10 levels, which may help modulate inflammation in the colonic tissue. Together, these effects contribute to the maintenance of epithelial barrier integrity, as evidenced by the preservation of tight junction protein expression.

## 4. Materials and Methods

### 4.1. Extraction of GOS from Lupinus albus

The extraction of galactooligosaccharides (GOS) from *Lupinus albus* was performed following the methodology described by Gulewicz [85]. Briefly, 100 g of defatted flour was extracted with 200 mL of 50% ethanol (*v*/*v*) at 40 °C under constant agitation overnight. The following day, the supernatant was collected by means of centrifugation at 3000 rpm for 15 min, and the flour was re-extracted with fresh 50% ethanol under the same conditions. The supernatants from both extraction cycles were combined and concentrated to a final volume of 25 mL using a rotatory vacuum evaporator (BUCHI, Essen, Germany) at 50 °C.

The concentrated extract was then gradually added to absolute ethanol under continuous stirring, maintaining a water extract to ethanol volume ratio of 1:10. The crude GOS precipitate was separated from the supernatant by means of centrifugation at 3000 rpm for 15 min. The GOS pellet was then placed into a vacuum desiccator overnight to remove any residual ethanol and water. The composition of GOS from *L. albus* was determined according to the official methods of analysis of AOAC International.

### 4.2. Mice

Seven-week-old C57BL/6 male mice purchased from the animal facility of the University Center for Health Sciences, University of Guadalajara, were housed under standard laboratory conditions (24 ± 1 °C, 12:12-h light/dark cycle), with ad libitum access to water and standard chow.

All experimental procedures were conducted following the protocols approved by the Research, Ethics, and Biosafety Committees of the University Center for Health Sciences, University of Guadalajara, under registration number CI-00420. This study was carried out in compliance with Ethical Guidelines for the Use of Animals in Research and the ARRIVE guidelines.

### 4.3. Experimental Design

After a one-week acclimation period, eight-week-old mice were randomly assigned to three experimental groups (*n* = 5 per group): Control (CTL), Ulcerative Colitis (UC), and Ulcerative Colitis with GOS (UC + GOS). Only animals with a body weight between 18 and 22 g were included in the randomization. A formal a priori power calculation for the sample size was not performed; instead, a minimal number of animals was selected to allow for valid statistical analysis while adhering to ethical principles of animal research.

The induction of ulcerative colitis was carried out by administering 2% dextran sulfate sodium (Sigma Aldrich, St. Louis, MO, USA, no. Cat 42867) in the mice’s drinking water for seven consecutive days. The UC + GOS group received *Lupinus albus*-derived galactooligosaccharides at a dose of 2.5 g/kg body weight, dissolved in sterile water and administered via oral gavage for 14 consecutive days. Meanwhile, the CTL and UC groups received sterile water to stimulate the gavage procedure of the UC + GOS group.

The GOS treatment protocol consisted of seven days of GOS administration prior to dextran sulfate sodium (DSS) exposure, followed by seven additional days of concurrent GOS and DSS administration (Figure 8).

Throughout the experimental period, the health status of each mouse was monitored daily. Animals that exhibited signs of severe distress or compromised activity—such as extreme lethargy, immobility, hunched posture, or inability to access food and water—were excluded from the study, in accordance with humane endpoint guidelines.

On the day following the final DSS administration, the mice were sacrificed. Prior to euthanasia, mice were anesthetized, and then euthanasia was performed by cervical dislocation.

The colon was carefully dissected from the cecum to the rectum, and mesenteric tissue was removed. Immediately after extraction, the colon length was measured. The descending colon was then divided into three segments. One fragment was fixed in 10% PBS-buffered formalin for histological analysis of colonic inflammation severity; a second fragment was fixed in Carnoy’s solution for mucus layer evaluation; and the third was rinsed with PBS and immediately snap-frozen in dry ice for subsequent RNA extraction.

For the cecum tissue, fecal contents were first collected, and the tissue was then rinsed with PBS and immediately snap-frozen for RNA extraction.

### 4.4. Assessment of Ulcerative Colitis Severity

During DSS administration, the disease activity index (DAI) was evaluated to monitor the progression of colitis. The DAI is a composite score determined by three clinical parameters: body weight loss, stool consistency, and the presence of rectal bleeding or blood in stools. The scoring was performed according to the criteria detailed in Table 1. Mice that did not reach a DAI score of ≥6 were excluded from the study.

To evaluate colonic inflammation, paraffin-embedded blocks of descending colon tissue fixed in 10% PBS-buffered formalin were sectioned into 5 μm-thick slices. For each sample, one slide containing five tissue sections was prepared. The sections were stained with hematoxylin and eosin (H&E, HYCEL, Zapopan, Mexico).

Colitis severity was assessed by evaluating key histopathological parameters, including inflammatory cell presence, leukocyte infiltration, fibrosis, epithelial damage, and mucosal damage. Each parameter was scored based on the criteria detailed in Table 2.

Histological evaluation was performed in a blinded manner by a professional pathologist.

### 4.5. Mucus Layer Measurement

For the mucus layer measurement, descending colon tissue was fixed in Carnoy’s solution (60% absolute ethanol, 30% chloroform, 10% glacial acetic acid, Sigma Aldrich, St. Louis, MO, USA) for 2.5 h without removing the fecal content. The tissue was then embedded in paraffin and sectioned into 3 μm-thick slices. Alcian Blue staining (HYCEL, Zapopan, Mexico), counterstained with Periodic Acid–Schiff (PAS, HYCEL, Zapopan, Mexico), was performed to visualize the mucus layer.

Mucus thickness was determined by measuring the distance between the epithelial surface and the outermost edge of the mucus layer using APERIO ImagenScope software 12.3.3 (LEICA, Illinois, IL, USA) at 20× magnification (scale bar: 100 μm) [31].

### 4.6. Feces Collection Protocol

Fecal samples were collected on days 0, 14, and 15 in all experimental groups. Samples from days 0 and 14 were obtained directly and placed into sterile tubes, and then immediately frozen in dry ice and stored at −80 °C until further analysis. On day 15, feces were extracted from the cecum following the measurement of colonic tissue. These samples were also collected in sterile tubes and immediately frozen in dry ice and stored at −80 °C until analysis.

### 4.7. Sample Preparation and DNA Sequencing

For each sampling time point- baseline (day 0), day 14, and cecum collection (day 15)—25 mg of fecal material was used for DNA extraction using the Quick-DNA™ Fecal/Soil Microbe Miniprep Kit (Zymo Research, Irvine, CA, USA, Cat. No. D6010) following the manufacturer’s instructions. DNA concentration and purity were assessed using a NanoDrop™ spectrophotometer (Thermo-Scientific, Waltham, MA, USA).

The hypervariable V3–V4 region of the 16S rRNA gene was amplified using primers and overhang adapter sequences according to the Illumina protocol. PCR reactions were carried out for each sample using the Platinum™ Taq High Fidelity DNA Polymerase (Invitrogen, Carlsbad, CA, USA, Cat No. 11304029). Amplified PCR products were purified using AMPure XP magnetic beads (Beckman Coulter, Brea, CA, USA, Cat. No. A63881), and DNA concentrations were quantified in duplicate using the Qubit™ dsDNA High Sensitivity Assay kit (Invitrogen, Carlsbad, CA, USA, Cat. No. Q32854).

Indexing and library preparation were performed according to the Illumina MiSeq System Protocol.

### 4.8. Bioinformatics Analysis and Statistics

A total of 7–8 mice per group were included in the sequencing analysis. Raw sequencing reads were obtained from the Illumina MiSeq System in FASTQ format and analyzed using QIIME 2 (version 2023.2). Amplicon sequence variants (ASVs) were generated using the DADA2 plugin and aligned against the SILVA rRNA database. An abundance table was constructed, containing ASVs and their corresponding counts per sample. ASVs that were unclassified at the bacterial kingdom level or classified as mitochondria or chloroplasts were excluded from downstream analysis. Alpha and beta diversity analyses were conducted based on the curated abundance table.

Statistical analyses were performed in R using RStudio (version 4.4.2, 31 October 2024). Alpha diversity metrics—including Observed Species (SOBs), Chao1, Shannon, Simpson, and Inverse Simpson indices—were calculated using the vegan package (version 2.6-8). These indices were compared using one-way ANOVA followed by Tukey’s post hoc test or the Kruskal–Wallis’s test followed by Dunn’s post-hoc test, depending on data distribution. For pre- and post-colitis comparisons, unpaired *t*-tests or Mann–Whitney U tests were applied, following assessment of data normality via the Shapiro–Wilk test.

Beta diversity was assessed using non-metric multidimensional scaling (NMDS) based on Bray–Curtis’s dissimilarity, also implemented in the vegan package. Differences in microbiota composition among experimental groups, time points, and sample sites were statistically evaluated using PERMANOVA with the adonis2 function. NMDS plots were generated using ggplot2 (version 3.5.1).

The ASV abundance table was transformed into a pivot table in Microsoft Excel 2024. Taxonomic abundance data (phyla, families, or genera) were extracted for specific time points, sampling sites, or experimental groups. The relative abundance of each taxon was calculated by dividing its read count by the total read count per sample (including unclassified reads). Relative abundances were visualized using violin plots in GraphPad Prism (version 9.5.0), with dashed and dotted lines representing the median and interquartile range, respectively. A *p*-value of ≤0.05 was considered statistically significant for all data analyses.

### 4.9. Real-Time RT-PCR

Total RNA was extracted using the TRizol™ reagent (Invitrogen, Carlsbad, CA, USA, Cat. No. 15596026) method. Complementary DNA (cDNA) was synthesized through reverse transcription of 2 mg of total RNA using the SuperScript™ II Reverse Transcriptase kit (Invitrogen, Carlsbad, CA, USA, Cat. No. 18064071), following the manufacturer’s instructions. Quantitative real-time PCR was performed using Maxima™ SYBR Green (Thermo Scientific, Waltham, CA, USA, Cat. No. K0251) on a StepOne Real-Time PCR System (Applied Biosystem, Waltham, CA, USA).

Gene expression analysis was conducted for monocarboxylate transporter 1 (MCT1), MCT4, occludin, claudin-1, claudin-2, claudin-4, junctional adhesion molecule A (JAM-A), zonula occludens 1 (ZO-1), interleukin-6 (IL-6), IL-10, IL-17, and tumor necrosis factor-alpha (TNF-α) using gene-specific forward and reverse primers. Each primer sequence was verified for alignment with its target gene using the NCBI BLAST program version 2.17.0. Primers sequences are listed in Appendix A [87,88,89,90,91,92,93,94].

Relative gene expression was calculated using the 2^ΔΔCT^ method and normalized to the expression of β-actin.

### 4.10. Statistical Analyses

All data are presented as the mean ± standard deviation (SD). Data normality was assessed using the Shapiro–Wilk test. A two-way ANOVA was used to evaluate the disease activity index (DAI), changes in body weight, and the expression levels of MCT1 and MCT4. Tukey’s multiple comparisons post hoc test was applied for DAI, body weight, and MCT4, while Sidák’s post hoc test was used for MCT1.

One-way ANOVA was performed to analyze the expression of claudin-2, JAM-A, ZO-1, and IL-10, as well as mucus layer thickness and colonic tissue length, followed by Tukey’s multiple comparisons post hoc test. One-way ANOVA was also used to evaluate the inflammatory score and the expression of TNF-α, IL-17, and IL-6, with Holm–Sídák post hoc test.

Non-parametric data, including the expression levels of claudin-1, claudin-4, and occludin, were analyzed using the Kruskal–Wallis test, followed by Dunn’s multiple comparisons post hoc test.

Statistical analyses were performed using the GraphPad Prism program (version 9.5.0, San Diego, CA, USA). A *p*-value ≤ 0.05 was considered statistically significant in all analyses.

## 5. Conclusions

Taken together, our findings suggest that GOS administration enhances intestinal health through multiple complementary mechanisms. GOS preserves the integrity of the mucus layer and supports the expression of tight junction proteins, contributing to the maintenance of the epithelial barrier. These effects were accompanied by selective modulation of the gut microbiota, including a reduction in potentially pro-inflammatory genera and an increase in beneficial taxa such as *Lactobacillus* and *Oribacterium*. In parallel, GOS regulated the expression of monocarboxylate transporters, potentially enhancing cellular metabolic activity and epithelial resilience by facilitating the transport of SCFAs into colonocytes.

Furthermore, GOS treatment attenuated colonic inflammation by downregulating pro-inflammatory cytokines while upregulating the anti-inflammatory cytokine IL-10.

Overall, these results highlight the potential of GOS as a targeted dietary intervention to reinforce gut barrier function and modulate immune responses in ulcerative colitis. Further studies are needed to elucidate the precise molecular pathways involved—particularly those related to inflammation signaling, epithelial transporter, and host microbiota interactions—and to assess the long-term preventive and therapeutic potential of GOS in inflammatory bowel diseases.

## Figures and Tables

**Figure 1 ijms-26-07968-f001:**
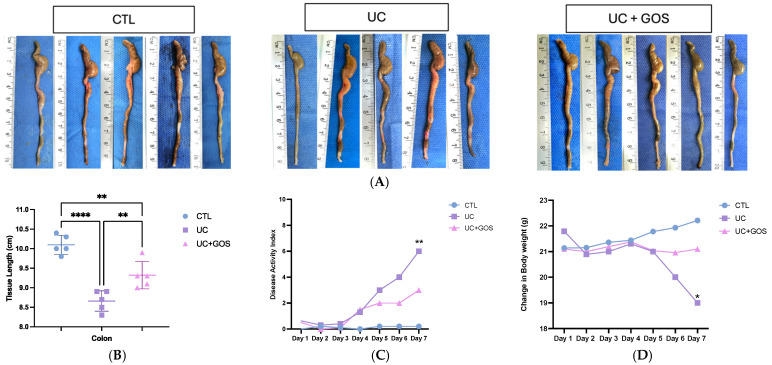
GOS treatment reduces tissue damage in a DSS-induced UC model. (**A**) Representative image of colonic tissue length. (**B**) Graphical representation of colonic tissue length. (**C**) DAI during DSS administration. (**D**) Body weight change during DSS administration. (**E**) Representative image of Alcian Blue staining with PAS counterstaining to evaluate mucus layer thickness, yellow arrows denote the mucus layer of the descending colon tissue (10× magnification; scale bar = 100 μm). (**F**) Representative H&E-stained sections showing colonic inflammation (10× magnification; scale bar = 100 μm). (**G**) Graphical representation of mucus layer thickness. (**H**) Inflammatory score quantification. Data are presented as the mean ± standard deviation (SD) for 5 mice per group. Statistical significance was determined using ordinary one-way ANOVA (**B**,**G**) with Tukey’s post hoc multiple comparisons test (** *p* ≤ 0.009; *** *p* ≤ 0.0003; **** *p* ≤ 0.0001). Ordinary one-way ANOVA (**H**) with Holm–Sidák’s multiple comparisons test (** *p* ≤ 0.006; **** *p* ≤ 0.0001). Two-way ANOVA (**C**,**D**) with Tukey’s multiple comparisons test (* *p* ≤ 0.01, ** *p* ≤ 0.009).

**Figure 2 ijms-26-07968-f002:**
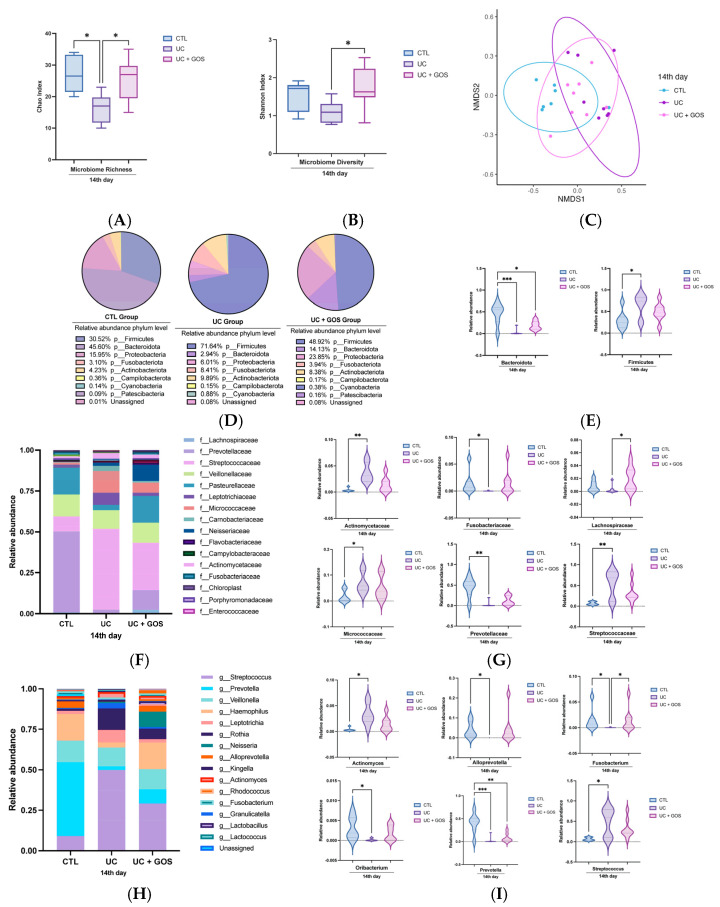
Comparison of gut microbiota composition among experimental groups on day 14. (**A**) Box plot representing the minimum, median, and maximum values of microbial richness. (**B**) Box plot representing the minimum, median, and maximum values of microbial diversity. (**C**) Non-metric multidimensional scaling (NMDS) analysis of gut microbiota composition across groups. Each point represents a sample, with colors indicating the corresponding group. (**D**) Relative abundance of the main bacterial phyla in each group, represented as a pie chart. (**E**) Violin plots showing significant changes at the phylum level. (**F**) Comparison of the relative abundance of key bacterial families among groups. (**G**) Violin plots depicting specific changes in bacterial families. (**H**) Comparison of the relative abundance of gut microbiota genera among groups. (**I**) Violin plots illustrating significant differences at the genus level. (* *p* ≤ 0.05, ** *p* ≤ 0.005, and *** *p* ≤ 0.0002). Data are presented as the median with 95% confidence intervals (**A**,**B**). Violin plots display median and quartile lines, along with the data distribution (**E**,**G**,**I**). A total of 7–8 mice per group were used for gut microbiota analysis.

**Figure 3 ijms-26-07968-f003:**
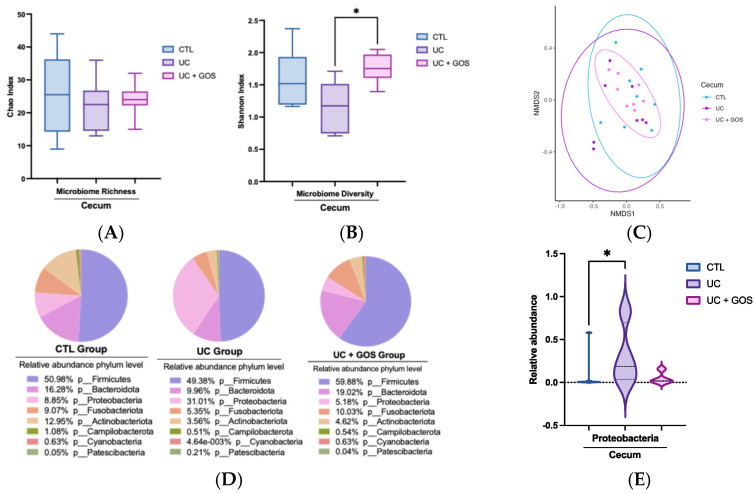
Comparison of gut microbiota composition among experimental groups in cecum feces. (**A**) Box plot representing the minimum, median, and maximum values of microbial richness. (**B**) Box plot representing the minimum, median, and maximum values of microbial diversity. (**C**) Non-metric multidimensional scaling (NMDS) analysis of gut microbiota composition across groups. Each point represents a sample, with colors indicating the corresponding group. (**D**) Relative abundance of the main bacterial phyla in each group, represented as a pie chart. (**E**) Violin plots showing significant changes at the phylum level. (**F**) Comparison of the relative abundance of key bacterial families among groups. (**G**) Comparison of the relative abundance of gut microbiota genera among groups (* *p* ≤ 0.05). Data are presented as the median with 95% confidence intervals (**A**,**B**). Violin plots display median and quartile lines, along with the data distribution (**E**). A total of 7–8 mice per group were used for gut microbiota analysis.

**Figure 4 ijms-26-07968-f004:**
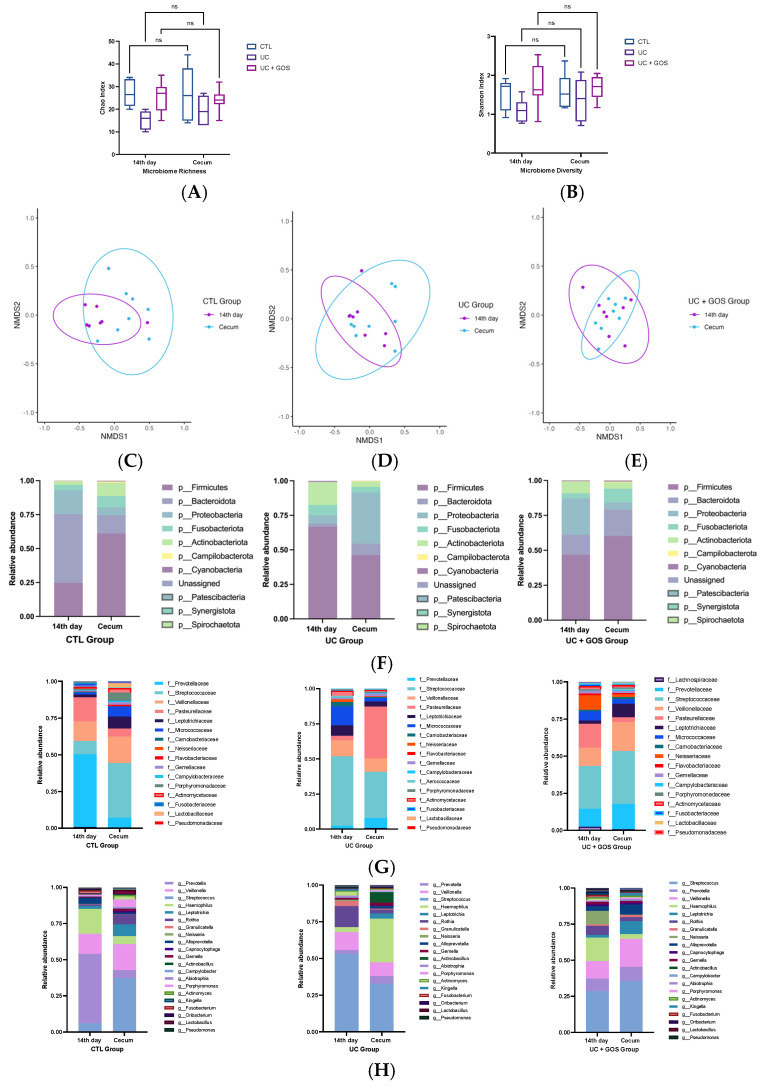
Comparison of gut microbiota composition between cecum and day 14 fecal samples. (**A**) Box plot representing the minimum, median, and maximum values of microbial richness. (**B**) Box plot representing the minimum, median, and maximum values of microbial diversity. (**C**) NMDS analysis comparing gut microbiota composition between day 14 and cecum feces in the CTL group. (**D**) NMDS analysis comparing gut microbiota composition between day 14 and cecum feces in the UC group. (**E**) NMDS analysis comparing gut microbiota composition between day 14 and cecum feces in the UC + GOS group. (**F**) Relative abundance of the main bacterial phyla in each group at day 14 and in cecum feces. (**G**) Comparison of the relative abundance of key bacterial families among groups at day 14 and cecum feces. (**H**) Comparison of the relative abundance of gut microbiota genera among groups at day 14 and in cecum feces. ns = non-significant differences. Data are presented as medians with 95% confidence intervals (**A**,**B**). A total of 7–8 mice per group were analyzed for gut microbiota composition.

**Figure 5 ijms-26-07968-f005:**
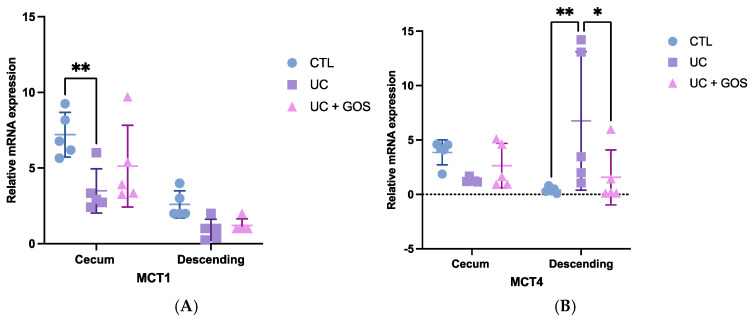
Expression of MCT1 and MCT4 in cecum and descending colon tissues. (**A**) Relative mRNA expression of MCT1 in cecum and descending colon tissues. (**B**) Relative mRNA expression of MCT4 in cecum and descending colon tissues. Data are presented as the mean ± standard deviation (SD); 5 mice per group were analyzed. Statistical significance was determined using two-way ANOVA (**A**) followed by Sidak’s multiple-comparisons test (** *p* ≤ 0.001). Two-way ANOVA (**B**) followed by Tukey’s multiple comparisons test (* *p* ≤ 0.027, ** *p* ≤ 0.008).

**Figure 6 ijms-26-07968-f006:**
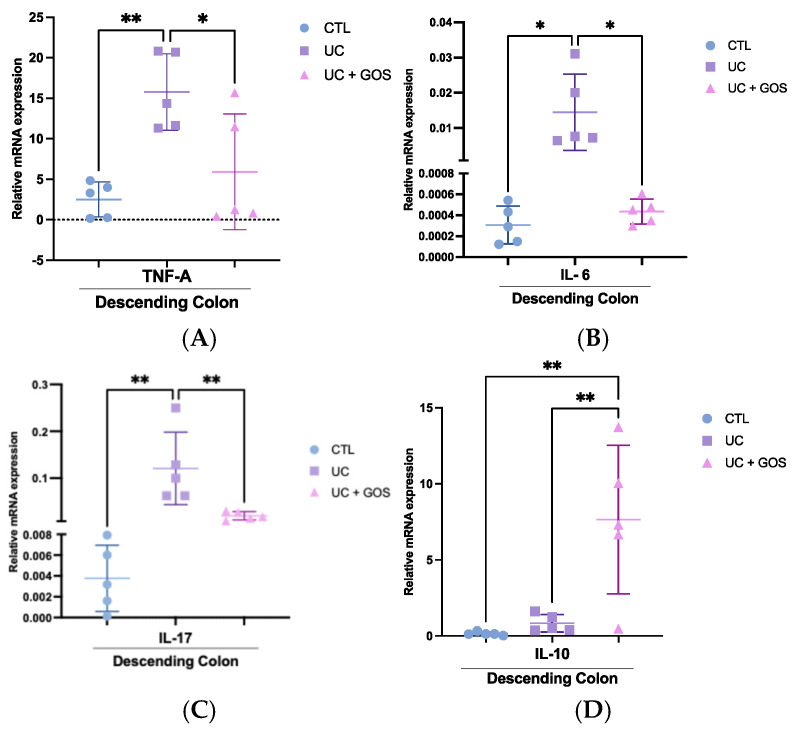
Alterations in cytokine expression in descending colonic tissue. (**A**) Relative mRNA expression of TNF-α in descending colon tissue. (**B**) Relative mRNA expression of IL-6 in descending colon tissue. (**C**) Relative mRNA expression of IL-17 in descending colon tissue. (**D**) Relative mRNA expression of IL-10 in descending colon tissue. Data are presented as the mean ± standard deviation (SD); 5 mice per group were analyzed. Statistical significance was determined using one-way ANOVA followed by Holm–Sídák’s multiple comparisons test ((**A**–**C**); ** *p* ≤ 0.005; * *p* ≤ 0.05). One-way ANOVA followed by Tukey’s multiple comparisons test was performed for the (**D**) (** *p* ≤ 0.006).

**Figure 7 ijms-26-07968-f007:**
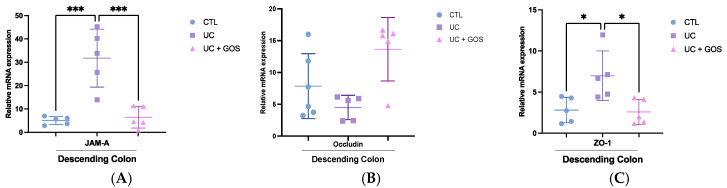
Tight junction mRNA expression in descending colonic tissue across experimental groups. (**A**) Relative mRNA expression of junctional adhesion molecule-A (JAM-A) in descending colon tissue. (**B**) Relative mRNA expression of occludin in descending colon tissue. (**C**) Relative mRNA expression of zonula ocludens-1 (ZO-1) in descending colon tissue. (**D**) Relative mRNA expression of claudin-1 in descending colon tissue. (**E**) Relative mRNA expression of claudin-2 in descending colon tissue. (**F**) Relative mRNA expression of claudin-4 in descending colon tissue. Data are presented as the mean ± standard deviation (SD); 5 mice per group were analyzed. Statistical significance was determined using one-way ANOVA followed by Tukey’s multiple-comparisons test for JAM-A, ZO-1, and claudin-2 (* *p* ≤ 0.02; ** *p* ≤ 0.001; *** *p* ≤ 0.0006). The Kruskal–Walli’s test followed by Dunn’s multiple comparisons test was used for occludin, claudin-1, and claudin-4 (* *p* ≤ 0.017).

**Figure 8 ijms-26-07968-f008:**
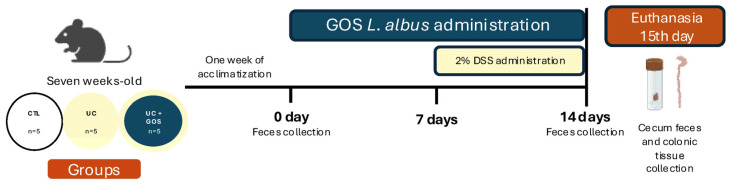
Schematic representation of the DSS-induced UC protocol and GOS treatment timeline.

**Table 1 ijms-26-07968-t001:** Disease activity index (DAI) scoring criteria.

Score	Weight Loss	Stool Consistency	Blood in Feces
0	0	Normal	Negative
1	1–5%	Soft but still formed	Trace amounts; blood steak
2	5–10%	Soft and unformed	Moderate bleeding; blood clot
3	10–20%	Loose	Visible, bloody stool
4	>20%	Diarrhea	Gross bleeding

Adapted from Godínez, L., et al. [27].

**Table 2 ijms-26-07968-t002:** Inflammatory score criteria for histological assessment of DSS-induced colitis.

Score	Presence of Inflammatory Cells	Extent of Leukocyte Infiltration	Severity of Fibrosis	Epithelial Damage	Mucosal Damage
0	Normal tissue	Rare (1–5 cells per field)	Normal	Normal	Normal tissue
1	Rare (1–5 per field)	Increased in lamina propria (5–10 cells)	Mild	Loss of the basal one-third of the crypt	1–2 foci of ulcerations
2	Slight increase (5–10 per field)	Confluent in the submucosal part	Moderate	Loss of the basal two-thirds of the crypt	2–4 foci of ulcerations
3	More obvious increase	Transmural infiltration	Severe	Entire crypt loss	Confluent or extensive ulceration
4	Significantly elevated (full infiltration)	-	Very Severe	Focal erosion	-
5	-	-		Confluent erosion	-

Adapted from Kwon J., et al. [86]. The symbol “-” indicates parameters not applicable for the corresponding category.

## Data Availability

Demultiplexed raw data are available in the Short Read Archive (SRA) under accession number PRJNA1276153.

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
