# Peer review of "Galactooligosaccharides Promote Gut Barrier Integrity and Exert Anti-Inflammatory Effects in DSS-Induced Colitis Through Microbiota Modulation"

_ijms, 2025, doi:10.3390/ijms26167968_

Round 1
Reviewer 1 Report
Comments and Suggestions for Authors
In some phrases it is difficult to grasp the idea that the authors wanted to convey. Perhaps it is necessary to put the idea in other words, perhaps the results or conclusions are not described correctly.
For example, the meaning stated in the paragraph (lines 154-162) is not clear. If there are differences in the composition of the microbiomes of the experimental groups before and after the intervention, but within each group the changes are not significant, what conclusions can be drawn? That the groups were initially different? "These findings suggest that while UC induction alters microbiota composition at the group level, individual variation within each group remains relatively stable" - It is not clear how this can happen if the experiment was performed correctly and adequate analysis methods were used?
The statement on lines 358-360 is not entirely clear – "Further, the observed increase in phyla Firmicutes in the UC group was more reflective of their basal abundance than a direct result of DSS administration." What did the authors mean by that?
A question about the use of terminology. Legend of Figure 3, line 230 et seq.: what is the term "cecal feces" - the contents of the cecum? Is it appropriate to use it in this case? Is it the same as "cecal samples" or something else? "Cecal feces" usually refers to the secretions of birds..
In general, I would like to improve the English language and the quality of the text.
And a few more technical notes:
Pie chart D in Figure 2 is empty;
Diagram H in Figure 4 is not drawn, empty
The numbering of the sections has been knocked down (line 313) section 2.5 or 2.7??
Paragraph (lines 397-406) it is not clear which works the discussion refers to. I remind you of the need to correctly post links to cited sources!
Lines 403-404 - it is not clear which parameter was increased and relative to which level, in which work.
In some phrases it is difficult to grasp the idea that the authors wanted to convey. Perhaps it is necessary to put the idea in other words, perhaps the results or conclusions are not described correctly.
For example, the meaning stated in the paragraph (lines 154-162) is not clear. If there are differences in the composition of the microbiomes of the experimental groups before and after the intervention, but within each group the changes are not significant, what conclusions can be drawn? That the groups were initially different? "These findings suggest that while UC induction alters microbiota composition at the group level, individual variation within each group remains relatively stable" - It is not clear how this can happen if the experiment was performed correctly and adequate analysis methods were used?
The statement on lines 358-360 is not entirely clear – "Further, the observed increase in phyla Firmicutes in the UC group was more reflective of their basal abundance than a direct result of DSS administration." What did the authors mean by that?
A question about the use of terminology. Legend of Figure 3, line 230 et seq.: what is the term "cecal feces" - the contents of the cecum? Is it appropriate to use it in this case? Is it the same as "cecal samples" or something else? "Cecal feces" usually refers to the secretions of birds..
In general, I would like to improve the English language and the quality of the text.
Author Response
In some phrases it is difficult to grasp the idea that the authors wanted to convey. Perhaps it is necessary to put the idea in other words, perhaps the results or conclusions are not described correctly.
Thank you for your valuable comment and for your time dedicated to reviewing our manuscript. We carefully revised the entire text to improve clarity and ensure that the ideas are conveyed in a more precise and understandable manner. We have rewritten several sections of the manuscript, especially in the results and discussion, to avoid potential confusion and to facilitate a better understanding of our findings and conclusions.
For example, the meaning stated in the paragraph (lines 154-162) is not clear. If there are differences in the composition of the microbiomes of the experimental groups before and after the intervention, but within each group the changes are not significant, what conclusions can be drawn?
That the groups were initially different?
"These findings suggest that while UC induction alters microbiota composition at the group level, individual variation within each group remains relatively stable" - It is not clear how this can happen if the experiment was performed correctly and adequate analysis methods were used?
We appreciate your detailed observation. We have rewritten the paragraph to clarify this point and improve the interpretation of the data.
We acknowledge that the sentence “These findings suggest that while UC induction alters microbiota composition at the group level, individual variation within each group remains relatively stable” was not appropriately phrased and may have led to confusion. We apologize for this and have removed it from the revised version of the manuscript.
Regarding the NMDS analysis results, we observed significant differences between experimental groups at day 14. However, when we analyzed the microbiota composition within each group— comparing their baseline (day 0) and day 14—we did not observe statistically significant differences. This suggests that the variation between groups at day 14 might be partly due to pre- existing differences in microbiota composition at baseline. Although the animals were randomized, individual microbial variability is known to exist even under controlled conditions, which could lead to group-level differences.
Additionally, we identified that the Firmicutes phylum showed a markedly higher relative abundance in the UC group at day 14 (71.64%) compared to CTL (30.52%) and UC + GOS (48.92%). This disproportionate representation largely explains the separation observed in the NMDS analysis. While other taxonomic differences were also statistically significant, their lower abundance reduced their impact on global beta-diversity visualization.
We also confirm that the statistical methods and analyses employed were appropriate and rigorously applied. The variability in this particular result is attributable to the aforementioned difference in phylum-level proportions.
These points have been clarified in the revised manuscript (lines 178-181 and 411-429)
The statement on lines 358-360 is not entirely clear – "Further, the observed increase in phyla Firmicutes in the UC group was more reflective of their basal abundance than a direct result of DSS administration." What did the authors mean by that?
Thank you for pointing out the lack of clarity in this statement. We intended to convey that the high proportion of Firmicutes observed in the UC group after DSS administration did not result from a significant increase relative to baseline but rather reflected its already elevated basal abundance in that group. When we compared the Firmicutes levels within the UC group between baseline and day 14, we did not observe a significant difference. Therefore, the high abundance detected post-colitis induction appears to be more attributable to its pre-existing concentration rather than to a specific effect of DSS treatment.
To avoid misunderstanding, we revised this sentence in the manuscript to more clearly express this finding and ensure alignment with the data presented (lines 426-429).
A question about the use of terminology. Legend of Figure 3, line 230 et seq.: what is the term "cecal feces" - the contents of the cecum? Is it appropriate to use it in this case? Is it the same as "cecal samples" or something else? "Cecal feces" usually refers to the secretions of birds.
Thank you for your insightful comment. In our manuscript, we used the term "cecal feces" to refer to the fecal content of the cecum. We found this term used in several publications as a synonym for "cecum feces" or "cecal content." However, we acknowledge that its use may confuse, particularly given its association with avian physiology. Considering your observation, we have standardized the terminology throughout the manuscript and now refer to these samples exclusively as "cecum feces" to ensure clarity and consistency.
•Choi SI, Son JH, Kim N, Kim YS, Nam RH, Park JH, Song CH, Yu JE, Lee DH, Yoon K, Min H, Kim YR, Seok YJ. Changes in Cecal Microbiota and Short-chain Fatty Acid During Lifespan of the Rat. J Neurogastroenterol Motil. 2021 Jan 30;27(1):134-146. doi:10.5056/jnm20148. PMID: 33380558; PMCID: PMC7786083.
In general, I would like to improve the English language and the quality of the text.
Thank you for your recommendation. The manuscript was reviewed and revised by a native English speaker affiliated with the University of Guadalajara to improve the overall language quality and ensure clarity throughout the text.
And a few more technical notes: Pie chart D in Figure 2 is empty;
Thank you for your observation. We believe this issue may have occurred during the file download or rendering process, as the original figure was correctly uploaded and does contain the three intended pie charts representing the phylum-level composition across the different experimental groups. We have double-checked the submission and confirmed that Figure 2D is complete in the version we uploaded. Nevertheless, we will ensure that the figure is visible and properly formatted in the final version.
Diagram H in Figure 4 is not drawn, empty
Thank you for your observation. Similar to the previous comment regarding Figure 2D, we believe this issue may have resulted from a problem during the file download. The original Figure 4H displays the relative abundance of gut microbiota of the different experimental groups at day 14 and the cecum, and it was properly included in the submitted manuscript.
The numbering of the sections has been knocked down (line 313), section 2.5 or 2.7??
The correct section number is 2.7. The error in the numbering was a typographical oversight that we unfortunately missed during revision. We appreciate your careful reading and bring our apologies for this mistake. The manuscript has been corrected accordingly. (line 334)
Paragraph (lines 397-406), it is not clear which work the discussion refers to. I remind you of the need to correctly post links to cited sources!
Thank you for your valuable observation. We have carefully revised and edited the paragraph to improve clarity regarding the discussed studies and to ensure proper citation formatting. We have ensured that all references are properly numbered and correspond to the cited works, facilitating clear traceability and accurate attribution. (line 460-471)
Lines 403-404 - it is not clear which parameter was increased and relative to which level, in which work.
Thank you for your comment. We have revised and edited the manuscript to clarify that the parameter increased is the relative abundance of the genus Haemophilus. Specifically, we detailed the comparison of Haemophilus levels after colitis induction (day 14) relative to baseline within the UC + GOS and CTL groups. (line 465-471)

Reviewer 2 Report
Comments and Suggestions for Authors
The manuscript was to elucidate the effect of GOS on gut microbiota modulation and the molecular mechanisms involved in epithelial restoration and inflammation reduction. Some suggestions are as followed:
- “5 rats per group” is a common starting point for experiments, but not a fixed criterion. Its adequacy depends on:
- Statistical power requirements (probability of detecting a true difference, needs to be ≥80%).
- Expected effect size (smaller effect requires more samples).
- Individual variability of the indicator (larger variability requires more samples).
- Pre-calculated sample sizes must be based on the above three points to ensure scientific rigor and ethical compliance (using a minimum number of valid numbers). Selection of 5 by convention alone is usually insufficient.
Authors are requested to provide a basis for compliance.
- The logic of the presentation of the introductory section is clear, but the organization of the content and the paragraph structure used by the author do not make sense, and it is recommended that the author rewrite the introduction.
- Abbreviations such as GOS\MCT-1 need to be added for the first time.
The authors' discussion still analyses the results of their own research, and it is recommended that more comparisons be made with current research to add to this section.
Author Response
The manuscript was to elucidate the effect of GOS on gut microbiota modulation and the molecular mechanisms involved in epithelial restoration and inflammation reduction. Some suggestions are as followed:
- “5 rats per group” is a common starting point for experiments, but not a fixed criterion. Its adequacy depends on:
- Statistical power requirements (probability of detecting a true difference, needs to be ≥80%).
- Expected effect size (smaller effect requires more samples).
- Individual variability of the indicator (larger variability requires more samples).
- Pre-calculated sample sizes must be based on the above three points to ensure scientific rigor and ethical compliance (using a minimum number of valid numbers). Selection of 5 by convention alone is usually insufficient.
Thank you for this valuable comment. We fully agree that the ideal sample size should be determined through prior statistical power analysis, taking into account expected effect size, variability, and desired significance level. In our study, the use of five mice per group was guided by ethical principles outlined in international, national, and institutional guidelines for animal experimentation, which emphasize the 3Rs—Reduction, Refinement, and Replacement—to minimize animal use while ensuring valid scientific outcomes.
Additionally, our choice was informed by several peer-reviewed studies investigating intestinal inflammation and microbiota modulation in similar DSS-induced colitis models, which also used five animals per group and demonstrated reproducible and interpretable results. Representative references:
- Park,H.;Yeo,S.;Kang,S.; Huh, C.S. Longitudinal Microbiome Analysis in a Dextran Sulfate Sodium-Induced Colitis Mouse Model. Microorganisms 2021, 9, 370. https://doi.org/10.3390/ microorganisms9020370
- Chassaing, B., Aitken, J. D., Malleshappa, M., & Vijay-Kumar, M. (2014). Dextran sulfate sodium (DSS)-induced colitis in mice. Current protocols in immunology, 104, 15.25.1–15.25.14. https://doi.org/10.1002/0471142735.im1525s104
- Yee, SM., Choi, H., Seon, JE. et al.Axl alleviates DSS-induced colitis by preventing dysbiosis of gut microbiota. Sci Rep 13, 5371 (2023). https://doi.org/10.1038/s41598-023-32527-2
Due to the exploratory nature of our work and the lack of previous effect size data for GOS from Lupinus albus, a formal power calculation was not feasible at this stage. Nonetheless, we acknowledge this limitation and plan to incorporate power analyses in future studies to ensure even greater statistical robustness.
Authors are requested to provide a basis for compliance.
2. The logic of the presentation of the introductory section is clear, but the organization of the content and the paragraph structure used by the author do not make sense, and it is recommended that the author rewrite the introduction.
Thank you for this helpful suggestion. Based on your recommendation, we have rewritten the introduction to improve the organization and coherence of the content. We refined the selection of background information and restructured the paragraphs to ensure a more logical flow and greater clarity in the narrative.
3. Abbreviations such as GOS\MCT-1 need to be added for the first time.
Thank you for pointing this out. We have revised the manuscript to ensure that all abbreviations, including GOS and MCT-1, are defined upon their first appearance in the text, specifically in the introduction, to avoid confusion. (line 86 and line 94)
4. The authors' discussion still analyses the results of their own research, and it is recommended that more comparisons be made with current research to add to this section.
We appreciate your recommendation. In response, we expanded the discussion to include additional comparisons with current literature. These comparisons help contextualize our findings concerning similar studies and strengthen the relevance and interpretation of our results. We hope these additions enhance the quality and depth of the discussion section.

Round 2
Reviewer 2 Report
Comments and Suggestions for Authors
ACCEPT
Comments on the Quality of English Language.